## Research Article

probable depression; anxiety disorder; women; household decision-making autonomy; Mozambique

**Corresponding author:**
Roger Antabe;
Email: roger.antabe@utoronto.ca

# Women's household decision-making autonomy and mental health outcomes in Mozambique

Roger Antabe[1] ⓘ, Gregory Antabe[2], Yujiro Sano[3] and Cornelius K. A. Pienaah[4]

[1]Department of Health and Society, University of Toronto Scarborough, Toronto, ON, Canada; [2]Department of Mathematics, Our Lady of Apostles School, Ahinsan Kumasi, Ashanti Region, Ghana; [3]Department of Sociology and Anthropology, Nipissing University, North Bay, ON, Canada and [4]Department of Geography, Western University, Social Science Centre, London, ON, Canada

## Abstract

Studies point to the role of sociocultural and household power dynamics in women's risk of mental illnesses. Using the context of Mozambique, we examined the association between women's household decision-making autonomy with probable depression and reporting symptoms of anxiety. We used the 2022–2023 Mozambique Demographic and Health Survey and applied logistic regression analysis. Our findings indicate high prevalence rates of depression (10%) and anxiety (11%) among married women. We also find that married women with the highest forms of household autonomy who take decisions alone on their health care (OR = 0.43, 95% CI = 0.32, 0.59; OR = 0.52, 95% CI = 0.38, 0.70), on making large household purchases (OR = 0.43, 95% CI = 0.28, 0.64; OR = 0.52, 95% CI = 0.35, 0.76) and visiting family members or relatives (OR = 0.36, 95% CI = 0.25, 0.51; OR = 0.64, 95% CI = 0.46, 0.89) were all less likely to report propable depression and symptoms of anxiety, respectively. Additionally, higher household wealth and employment acted as protective assets against both depression and anxiety. We recommend working to remove the sociocultural barriers to women's autonomy while improving their socioeconomic status, such as income and employment opportunities, which will lead to a better mental health outcome and serve as an important pathway to increasing their autonomy.

## Impact statement

Research has suggested that women are exposed to an increased risk of mental health, including probable depression and anxiety, relative to men. This is particularly the case of Mozambique, where previous studies have discussed the over-exposure of women to depression and anxiety. Studies have unpacked the contributory factors for this increased risk to include biological mechanisms, including the role of reproductive hormones. Others have also discussed sociocultural factors and household power dynamics in women's increased susceptibility to depression and anxiety. Using the context of Mozambique, our study examined the association of women's household decision-making autonomy with reporting probable depression and symptoms of anxiety. Our findings suggest that women with higher household decision-making autonomy in the country are less likely to report depression and symptoms of anxiety relative to those with the lowest levels of autonomy. Our findings make an important contribution to mental health policy in Mozambique and elsewhere in Sub-Saharan Africa. We recommend improving women's autonomy as part of a holistic strategy to improve their mental health outcomes.

## Introduction

In Mozambique, the prevalence of depression and anxiety are not only endemic but tend to be highly gendered. Relative to their male counterparts, women have a higher likelihood of reporting or being diagnosed with depression and anxiety. Among people using outpatient psychiatric services in the country, more women (12.8%) were diagnosed with neurotic disorders, which are inclusive of depression and anxiety, relative to their male counterparts (5.7%) (Pires et al., 2019). This pattern is worse for female-headed households in the country, where an earlier study found a 14% prevalence rate for depression (Audet et al., 2018). Among three groups of women in two semi-urban settings in Mozambique, Khan et al. (2022) found that the prevalence of any common mental disorders, such as major depressive episodes and generalized anxiety disorder, ranged from 35.3% to 40.7%. Confirming similar findings in the Sofala Province of the country, Wagenaar et al. (2016) also found that in the outpatient psychiatric visits, women seeking treatment were more likely to be for mood and neurotic disorders, such as depression and anxiety, than for substance use and epilepsy, which were found to be more common among men.

Unpacking the higher rates of depression and anxiety among women, studies have discussed both biological mechanisms and social processes to underpin their relative risk. For instance, fluctuating reproductive hormones like the levels of estrogen are noted to underpin and heighten women's susceptibility to depression and anxiety compared to their male counterparts (Soares and Zitek, 2008; Kundakovic and Rocks, 2022). The cerebral structures, neural correlates, immune system and inflammatory reaction have also been noted as important biological mechanisms accounting for the observed differences with men (Di Benedetto et al., 2024). Studies hinting at the influence of sociocultural factors have discussed issues such as the reproductive pressure on women, such as child rearing, lack of support and low socioeconomic status (Kadri and Alami, 2009). Importantly, scholars have emphasized the socialization process where the responsibilities and expectations placed on women contribute to inducing stress and anxiety among them (Salk et al., 2017).

In the settings of Mozambique, Khan et al. (2022) observed that women's heightened risk of depression and anxiety were linked to coercive reproduction, their high prevalence of HIV, low acceptance of women in the workforce and low educational attainment. The authors further noted that women of childbearing age in the country are exposed to other sociocultural pressures, such as gender inequality and cultural expectations concerning women's role in society. Others included intimate partner violence and women's submission to cultural gender roles and duties. All of these worked to heighten their risk of depression and anxiety compared to men. Thus, women's sociocultural context and household power dynamics work directly and indirectly to expose them to depression and anxiety (Astbury, 2010; Kuruvilla and Jacob, 2019). Regardless of this observation, however, there is no study in Mozambique examining how women's household decision-making autonomy (which is mainly underpinned by sociocultural values and expectations) is associated with their risk of depression and anxiety. This is particularly concerning as women in Mozambique are observed to possess lower scores of decision-making autonomy (UN Women, 2019). In the patriarchal settings of SSA, studies have alluded to the influence of the sociocultural environment on married women's poor mental health outcomes, including depression and anxiety (Kaplan, 2021, 2023). The specific objective of this study is to examine how the household power dynamics, which is reflected in married women's decision-making autonomy contributes to their risk of poor mental health. The focus on married women is premised on the sociocultural dynamics of marriage in Mozambique and elsewhere in SSA, where they may experience heightened challenges making independent decisions regarding their health (Sano et al., 2017; Zegeye et al., 2023). Our study, therefore, makes an important contribution to the health policy and literature in Mozambique by examining the association between women's household decision-making autonomy and their risk of reporting probable depression and symptoms of anxiety. The findings will inform health policy on the influence of contextual sociocultural dynamics on married women's mental health outcomes, specifically depression and anxiety in Mozambique and similar contexts in SSA.

While women's empowerment may signal aspects such as their socioeconomic status, including education, wealth and employment, their household decision-making autonomy may connote the complex gender power dynamics within the household that position them to make independent decisions about their lives (Anfaara et al., 2024; Mason, 1986; Sano et al., 2018). Specifically, women's household decision-making autonomy constitutes women's ability to independently make decisions in the household regarding their own lives without the interference of or permission from male partners (Anfaara et al., 2024). The sociocultural settings tend to shape gender expectations, responsibilities and household power dynamics that may directly and/or indirectly impact their household decision-making autonomy (Acharya et al., 2010; Fuseini et al., 2019). These revelations reflect the lived realities of women in Mozambique, where evidence has suggested their low levels of autonomy in all major aspects of decision-making, including sexual, reproductive health, financial and economic (Intalian Agency for Development Cooperation, 2018; UN Women, 2019; Castro Lopes et al., 2024). Regarding household decision-making autonomy, Luz and Agadjanian (2015) found that 39% of women in four districts of the Gaza Province in Southern Mozambique had low levels of autonomy. Similarly, among women living with the human immunodeficiency virus (HIV) in the country, only 39% indicated higher household decision-making autonomy (Parcesepe et al., 2020).

Women's household decision-making autonomy is associated with many desirable health behaviours and health outcomes. For instance, across SSA, Zegeye et al. (2023) found that married women with higher levels of household decision-making autonomy were more likely to enrol in health insurance. Similarly, Seidu et al. (2021) established that compared with women with low household decision-making autonomy, those with medium and high levels of autonomy were more likely to negotiate for safer sex with their male partners across 27 countries in SSA. In Ghana, it was also found that women reporting higher scores of household decision-making autonomy on their health were more inclined to use antenatal care services (Ameyaw et al., 2016). Furthermore, research findings from Malawi have also suggested that women with higher household decision-making autonomy were more knowledgeable about HIV transmission relative to their counterparts with low levels of autonomy (Antabe et al., 2020).

Other positive health outcomes associated with higher levels of household decision-making autonomy include women's improved health and well-being with reduced exposure to all forms of intimate partner violence (Annan et al., 2021; Kebede et al., 2021). On the subject of depression, it was established among women in rural Burkina Faso that those with higher levels of empowerment – measured with some parameters on their household decision-making autonomy, self-efficacy and mutual respect among household members – were less likely to report stress and depression (Leight et al., 2022). In Senegal, Fielding and Lepine (2017) also found that women's higher empowerment, measured by their household decision autonomy, was negatively associated with anxiety and depression. Regardless of this evidence and the revelation that women in Mozambique may have lower levels of autonomy, no study has examined its association with their heightened prevalence rates of depression and anxiety. Our study, therefore, aims to contribute to the literature and health policy in Mozambique by examining the association between women's household decision-making autonomy and their risk of reporting probable depression and symptoms of anxiety.

## Methods

### Data and procedure

The study utilized secondary data from the Mozambique Demographic and Health Survey (MDHS) conducted from July 2022 to March 2023. The MDHS is a nationwide survey that is collected by the Instituto Nacional de Estatística, employing standardized

questionnaires and a multi-stage stratified cluster sampling method. This method included stratification by province and urban/rural areas, with equal probability sampling for primary sampling units and proportional sampling for enumeration areas. Specifically, the sampling framework followed a two-stage approach, designed to generate representative data at the national level, as well as for urban and rural regions and each of the 10 provinces. In the initial stage, clusters were chosen based on enumeration areas (EAs). A total of 613 EAs were identified using a probability proportional to size method, where the number of households in each stratum determined the size. Out of these, 230 clusters came from urban settings, while 383 were from rural locations. During the second stage, 26 households were systematically selected from each EA, ensuring equal selection probability. The MDHS provides high quality and reliable information on basic demographic indices and mental health, such as PHQ-9 and GAD-7. The interviews were completed with 13,183 women aged 15–49 and 5,380 men aged 15–54, with a response rate of 94% and 86% for women and men, respectively. In the MDHS, women's autonomy was only asked among women who are married and living with their partners, thereby excluding unmarried, divorced and widowed women. To this end, we focused on the weighted sample of 8,488 women as part of our analytical sample for this study. The MDHS was approved by the Instituto Nacional de Estatística Maputo, Moçambique and the Institutional Review Board of ICF, Rockville, Maryland, USA. Interviewers obtained informed consent by reading the consent statement of the respondents, who may accept or decline to participate.

### Measures

In our analysis, we have utilized the PHQ-9 and GAD-7 as dependent variables to assess mental health symptoms related to depression and anxiety. The PHQ-9 is a widely used self-report scale to measure the severity of depression, and it consists of 9 items rated on a 4-point scale, with strong psychometric properties and robust internal consistency. Relatedly, the GAD-7 measures the severity of generalized anxiety disorder symptoms with 7 items rated on a scale from 0 to 3 and demonstrates robust internal consistency. To streamline the data analysis and enhance interpretability, we created a binary variable derived from these scores. Specifically, for the PHQ-9, scores were categorized into depressed (indicating moderate to severe depression with scores of 10 or above) and not depressed (scores below 10). Similarly, the GAD-7 scores were classified into anxious (indicating moderate to severe anxiety with scores of 10 or above) and not anxious (scores below 10). This binary classification allows for a clear differentiation between people experiencing significant symptoms and those who are not, facilitating a more focused examination of the associations and impacts of depression and anxiety in our study population. In addition, women were asked, 'Who usually has the final say in household settings on the following decisions: 1) obtaining their own health care, 2) making large purchases, and 3) visits to family and relatives. These variables had three response categories (0 = respondent only; 1 = respondent and partner; 2 = partner/other). These variables were adopted as our independent variables. Moreover, to account for potential confounding effects, we controlled for a range of sociodemographic characteristics, such as education, household wealth, employment, age, place of residence, religion, marital status and total children ever born.

### Statistical analysis

We employed descriptive analysis to show sample characteristics. In addition, we used unadjusted and adjusted logistic regression analysis to estimate the association between women's autonomy and two types of mental health indicators – the PHQ-9 and GAD-7. We use odds ratios (ORs) to present the results, where values exceeding 1 suggest a higher likelihood of reporting PHQ-9/GAD-7 scores of 10 or higher, and values below 1 indicate reduced odds of reporting these scores. All analyses were conducted using STATA 17 (State Corp, College Station, TX, USA). The "svy" function was applied during the statistical analysis to account for the cluster sampling design and sampling weights.

### Results

Table 1 shows descriptive findings. Most respondents demonstrate low levels of depression and anxiety, with 90% scoring below 10 on the PHQ-9 and 89% on the GAD-7. In health care decisions, 30% of respondents rely on their partners or others, while 40% make decisions jointly with their partners. This trend is similarly reflected in large household purchases, where 43% of respondents defer to their partners or others, and 44% make decisions collaboratively. For visits to family or relatives, the pattern continues, with 33% relying on their partners and 45% sharing decision-making responsibilities. Educational attainment is relatively limited, with 45% of participants having only primary education and 32% lacking formal education altogether. In terms of socioeconomic status, the household wealth distribution is relatively even across different categories, but a significant 71% of the sample is unemployed. In addition, the sample is predominantly rural (68%).

Table 2 shows findings from logit models predicting PHQ-9. In Model 1, which focuses on health care decisions, women who make decisions independently (respondent alone) show significantly lower odds of depression (OR = 0.33, $p < 0.001$) compared to those who rely on partners or others, while joint decision-making (respondent and partner) offers some protection (OR = 0.70, $p < 0.05$). In Model 2, once we accounted for sociodemographic variables, the impact of joint decision-making on mental health becomes no longer significant (OR = 0.88, $p > 0.05$); however, women who make decisions independently still show significantly lower odds of depression (OR = 0.43, $p < 0.001$) compared to those who rely on partners or others. Model 3, which examines decision-making related to large household purchases, reveals that women making decisions independently have significantly lower odds of depression (OR = 0.34, $p < 0.001$) compared to those who depend on partners or others. Joint decision-making also shows a less significant protective effect (OR = 0.65, $p < 0.01$). In Model 4, the analysis incorporates additional sociodemographic variables, reinforcing the previous patterns. Independent decision-making maintains a significant association with lower odds of depression (OR = 0.43, $p < 0.001$), while joint decision-making does not demonstrate a significant effect (OR = 0.78, $p > 0.05$). Turning to decision-making related to visits to family or relatives, as shown in Model 5, independent decision-making remains linked to lower odds of depression (OR = 0.29, $p < 0.001$), while joint decision-making shows a less significant protective effect (OR = 0.73, $p < 0.05$). In a fully adjusted model (Model 6), we find that independent decision-making continues to maintain its significant association with lower odds of depression (OR = 0.36, $p < 0.001$), while joint decision-making does not demonstrate a significant effect (OR = 0.87, $p > 0.05$). In addition to women's autonomy, we found a range of control variables associated with depression; for instance, richer women are generally less likely to report depression compared to their poorer counterparts, and employed women were less likely to report depression compared to unemployed women. Older women

**Table 1.** Weighted sample characteristics

| | Percentage | Weighted counts |
|---|---|---|
| **PHQ–9** | | |
| Less than 10 | 90 | 7,620 |
| 10 or greater | 10 | 867 |
| **GAD–7** | | |
| Less than 10 | 89 | 7,515 |
| 10 or greater | 11 | 972 |
| **Person who usually decides on: respondent's health care** | | |
| Partner/other | 30 | 2,577 |
| Respondent and partner | 40 | 3,370 |
| Respondent alone | 30 | 2,541 |
| **Person who usually decides on: large household purchases** | | |
| Partner/other | 43 | 3,645 |
| Respondent and partner | 44 | 3,729 |
| Respondent alone | 13 | 1,113 |
| **Person who usually decides on: visits to family or relatives** | | |
| Partner/other | 33 | 2,769 |
| Respondent and partner | 45 | 3,849 |
| Respondent alone | 22 | 1,870 |
| **Education** | | |
| Secondary education or higher | 23 | 1,919 |
| No education | 32 | 2,712 |
| Primary education | 45 | 3,857 |
| **Household wealth** | | |
| Poorer | 20 | 1,711 |
| Poorest | 21 | 1,804 |
| Middle | 20 | 1,705 |
| Richer | 20 | 1,654 |
| Richest | 19 | 1,613 |
| **Employment** | | |
| No | 71 | 5,998 |
| Yes | 29 | 2,490 |
| **Age** | | |
| 15–19 | 11 | 951 |
| 20–24 | 21 | 1,823 |
| 25–29 | 20 | 1,737 |
| 30–34 | 15 | 1,256 |
| 35–39 | 13 | 1,122 |
| 40–44 | 10 | 857 |
| 4,549 | 9 | 742 |
| **Place of residence** | | |
| Urban | 32 | 2,735 |

*(Continued)*

**Table 1.** *(Continued)*

| | Percentage | Weighted counts |
|---|---|---|
| Rural | 68 | 5,753 |
| **Religion** | | |
| Muslim | 22 | 1,887 |
| Catholic | 30 | 2,571 |
| Zion | 12 | 1,038 |
| Evangelical/Pentecostal | 26 | 2,203 |
| No religion | 7 | 610 |
| Other | 2 | 178 |
| **Martial status** | | |
| Married | 43 | 3,660 |
| Living with partner | 57 | 4,827 |
| **Total children ever born** | | |
| None | 7 | 632 |
| One | 17 | 1,423 |
| Two | 19 | 1,608 |
| Three | 17 | 1,428 |
| Four or more | 40 | 3,397 |
| Total | 100 | 8,488 |

are also generally more likely to report depression compared to their youngest counterparts. Furthermore, women who belong to Zion and Evangelical/Pentecostal faiths were both less likely to report depression compared to their Muslim counterparts. Interestingly, women who have given birth to four or more children were less likely to report depression compared to women who have never given birth.

Table 3 shows findings from logit models predicting GAD-7. In Model 1, which examines health care decisions, women making decisions independently (respondent alone) show significantly lower odds of anxiety (OR = 0.41, $p < 0.001$) compared to those who depend on partners or others. Further adjusting for sociodemographic factors in Model 2, independent decision-makers continue to show lower odds of anxiety (OR = 0.52, $p < 0.001$). In Model 3, related to visits to family or relatives, women who make decisions alone continue to demonstrate significantly lower odds of anxiety (OR = 0.43, $p < 0.001$), while joint decision-making shows a less significant protective effect (OR = 0.73, $p < 0.05$). In Model 4, with further adjustment for sociodemographic factors, independent decision-making maintains its strong association with lower anxiety (OR = 0.52, $p < 0.001$), whereas joint decision-making remains non-significant (OR = 0.81, $p > 0.05$). In Model 5 and Model 6, the protective effect of independent decision-making persists across contexts, with OR of 0.53 ($p < 0.001$) and 0.64 ($p < 0.001$) in an unadjusted and adjusted model, respectively. Beyond women's autonomy, we found a range of control variables associated with anxiety; for instance, richer women are generally less likely to report anxiety compared to their poorer counterparts. Older women are also generally more likely to report anxiety compared to their youngest counterparts. Furthermore, women who belong to Zion and Evangelical/Pentecostal faiths were both less likely to report anxiety compared to their Muslim counterparts.

Interestingly, women who have given birth to four or more children were less likely to report anxiety compared to women who have never given birth.

## Discussion

Our findings revealed high prevalence rates of depression and anxiety among women in the country. Specifically, there is a 10% and 11% prevalence rates for depression and anxiety, respectively. While these reported prevalence rates are relatively higher compared to the global rates of 3.6% and 4.4% for depression and anxiety, they are consistent with that of the SSA region at 9% for depression and 10% for anxiety (WHO, 2017). However, we note that the rates found by our study are lower than what earlier studies had projected in Mozambique, implying the government's strategy on addressing mental health through the Programa de saúde mental 2006–2015 policy document may have yielded some positive results in the country. Indeed, Halsted et al. (2019) acknowledged the success of mental health strategies in the country, where the majority of those reporting mental health issues such as depression and anxiety get the needed care.

Our findings suggest that women in Mozambique who exercise the highest level of household decision-making autonomy, that is, making decisions alone regarding their health care, making large household purchases and visiting family members, were all less likely to report probable depression and symptoms of anxiety. These findings shed additional critical insight into the role of Mozambican women's household decision-making autonomy in their exposure to poor mental health, particularly depression and anxiety. The association of women's low autonomy with the heightened risk of depression and anxiety unveils the potential underlying influence of cultural pressures working through social structures and household power dynamics in Mozambique. This revelation contributes to the emerging discussion by scholars such as Astbury (2010), alluding to the need for research and policymakers not only to examine how women's poor mental health is influenced by sociocultural dynamics but also the urgency in using a human rights framework to mitigate women's poor mental health outcomes. An earlier study in the country by Khan et al. (2022) signalled the role of sociocultural dynamics, including gender inequality, cultural expectations about women's role in society, intimate partner violence, and women's cultural submission to be associated with their exposure to poor mental health, including depression and anxiety. Hinged on this premise, we argue that women in Mozambique who are exposed to the worst forms of these adverse sociocultural dynamics will experience the lowest form of autonomy, which our study has observed to be associated with a heightened risk of reporting depression and symptoms of anxiety relative to women with higher autonomy. The observation in this study is also consistent with that of previous studies that have established the protective effect of women's improved autonomy for adverse health outcomes, particularly in relation to sexual and reproductive health (Kamiya, 2011; Ameyaw et al., 2016; Antabe et al., 2020; Seidu et al., 2021).

We also observed the role of women's socioeconomic characteristics on their risk of depression and reporting symptoms of anxiety. For instance, women belonging to wealthier households relative to those in the poorest households were less likely to have depression and anxiety. This finding is explained by the revelation from earlier studies suggesting that improved household wealth acted as a protective factor against poor health outcomes, including

mental health. Contextualizing a similar finding, Liu et al. (2023) observed in Europe that higher household income was associated with a lowered risk for genetic liability for depression and anxiety disorders. Similarly, in a scoping review, Ettman et al. (2022) found an inverse relationship between household wealth and depression, observing that wealth status influenced depression along the life course, wealth protected against depression in the face of stressors and savings worked to reduce depression. Again, employed women, compared to their unemployed counterparts, were less likely to report depression. This may be explained by the stress and worry associated with an inability to meet financial responsibilities and basic needs, which may work to induce depression and anxiety (Arena et al., 2023). In a systematic review, Amiri (2022) found that unemployment increased the risk of depression, explaining one potential pathway to include the likelihood of the unemployed engaging in risky, unhealthy behaviours and lifestyles that may compromise their mental health. A study in India among youths also established similar findings where depression and anxiety were higher among the unemployed (Biswas et al., 2024).

Some demographic features of women were also associated with their risk of depression and anxiety. We noted that compared to the youngest age cohort, older age groups of women were more likely to report depression and anxiety. This finding is consistent with the observation by the World Health Organization (2017), positing that increasing age was a risk factor for mental illnesses such as depression and anxiety. Explaining the relationship between age, depression and anxiety, other scholars have indicated increasing age is associated with major life events and health conditions that may induce depression and anxiety relative to younger people (Gao et al., 2023; Graham et al., 2024). We further found that compared to Muslim women, all other Christian denominations, those without religion and those in other religions were all less likely to report both depression and anxiety. While this finding calls for additional research insights into the role of religious affiliation and mental health outcomes, some earlier studies elsewhere have noted that a Christian religious affiliation may be associated with some coping mechanism that works to reduce their exposure to poor mental health (Gray and Cook, 2021; Li et al., 2024).

Finally, it emerged from our findings that higher birth parity was associated with depression and anxiety. We observed that women with four or more children were less likely to report depression and anxiety relative to their colleagues without children. This finding contrasts with Khan et al. (2022) that suggested that because women in Mozambique received little to no support for child rearing, this contributed to worsen their mental health depression and anxiety relative to men. We, however, note that this earlier finding, unlike our study, was not nationally representative and was indeed limited to some few women attending a health facility. We argue that in a highly patriarchal and pronatalist society such as Mozambique, married women without children may be more vulnerable to anxiety and depression. This is because they tend to be stigmatized and socially isolated, a phenomenon that works to deteriorate their mental health.

We have some noteworthy limitations to our study. The cross-sectional nature of the MDHS limits our findings to statistical association. We also note that the measures of depression and anxiety are through self-reported responses. This makes them susceptible to recall bias from participants. Furthermore, we note that focusing our study sample on only currently married women excludes the applicability and implications of our findings to a diverse group of women in the country, including divorced, separated and never married women. Lastly, our construct of women's

**Table 2.** Logit models predicting PHQ-9 among married women in Mozambique

| | Respondent's health care | | | | | | Large household purchases | | | | | | Visits to family or relatives | | | | | |
|---|---|---|---|---|---|---|---|---|---|---|---|---|---|---|---|---|---|---|
| | Model 1 | | | Model 2 | | | Model 3 | | | Model 4 | | | Model 5 | | | Model 6 | | |
| | OR | 95% CI | | OR | 95% CI | | OR | 95% CI | | OR | 95% CI | | OR | 95% CI | | OR | 95% CI | |
| **Autonomy** | | | | | | | | | | | | | | | | | | |
| Partner/other | 1.00 | | | 1.00 | | | 1.00 | | | 1.00 | | | 1.00 | | | 1.00 | | |
| Respondent and partner | 0.70* | 0.52 | 0.95 | 0.88 | 0.64 | 1.22 | 0.65** | 0.49 | 0.87 | 0.78 | 0.58 | 1.04 | 0.73* | 0.55 | 0.99 | 0.87 | 0.63 | 1.19 |
| Respondent alone | 0.33*** | 0.24 | 0.45 | 0.43*** | 0.32 | 0.59 | 0.34*** | 0.22 | 0.50 | 0.43*** | 0.28 | 0.64 | 0.29*** | 0.20 | 0.42 | 0.36*** | 0.25 | 0.51 |
| **Education** | | | | | | | | | | | | | | | | | | |
| Secondary education or higher | | | | 1.00 | | | | | | 1.00 | | | | | | 1.00 | | |
| No education | | | | 1.03 | 0.73 | 1.45 | | | | 1.02 | 0.72 | 1.43 | | | | 1.04 | 0.74 | 1.46 |
| Primary education | | | | 1.31 | 1.00 | 1.71 | | | | 1.30 | 0.99 | 1.70 | | | | 1.34* | 1.03 | 1.75 |
| **Household wealth** | | | | | | | | | | | | | | | | | | |
| Poorer | | | | 1.00 | | | | | | 1.00 | | | | | | 1.00 | | |
| Poorest | | | | 0.68* | 0.48 | 0.95 | | | | 0.69* | 0.50 | 0.96 | | | | 0.68* | 0.49 | 0.96 |
| Middle | | | | 0.53** | 0.35 | 0.79 | | | | 0.54** | 0.36 | 0.80 | | | | 0.53** | 0.36 | 0.79 |
| Richer | | | | 0.64* | 0.42 | 0.97 | | | | 0.66* | 0.43 | 0.99 | | | | 0.64* | 0.42 | 0.97 |
| Richest | | | | 0.34*** | 0.20 | 0.57 | | | | 0.35*** | 0.21 | 0.60 | | | | 0.32*** | 0.19 | 0.55 |
| **Employment** | | | | | | | | | | | | | | | | | | |
| No | | | | 1.00 | | | | | | 1.00 | | | | | | 1.00 | | |
| Yes | | | | 0.71* | 0.53 | 0.94 | | | | 0.70* | 0.53 | 0.92 | | | | 0.71* | 0.54 | 0.95 |
| **Age** | | | | | | | | | | | | | | | | | | |
| 15–19 | | | | 1.00 | | | | | | 1.00 | | | | | | 1.00 | | |
| 20–24 | | | | 1.45 | 0.97 | 2.17 | | | | 1.48 | 0.99 | 2.20 | | | | 1.47 | 0.98 | 2.19 |
| 25–29 | | | | 1.60* | 1.01 | 2.54 | | | | 1.65* | 1.04 | 2.62 | | | | 1.64* | 1.03 | 2.62 |
| 30–34 | | | | 1.89* | 1.13 | 3.16 | | | | 1.98** | 1.18 | 3.30 | | | | 1.92* | 1.15 | 3.22 |
| 35–39 | | | | 2.29*** | 1.42 | 3.69 | | | | 2.37*** | 1.47 | 3.83 | | | | 2.33*** | 1.44 | 3.77 |
| 40–44 | | | | 1.51*** | 0.92 | 2.49 | | | | 1.63 | 0.98 | 2.69 | | | | 1.57 | 0.96 | 2.57 |
| 45–49 | | | | 2.34 | 1.44 | 3.80 | | | | 2.46*** | 1.51 | 4.02 | | | | 2.42*** | 1.49 | 3.95 |
| **Place of residence** | | | | | | | | | | | | | | | | | | |
| Urban | | | | 1.00 | | | | | | 1.00 | | | | | | 1.00 | | |
| Rural | | | | 0.64* | 0.42 | 0.98 | | | | 0.65* | 0.42 | 0.99 | | | | 0.65 | 0.43 | 0.98 |

(*Continued*)

| | Respondent's health care | | | | Large household purchases | | | | Visits to family or relatives | | | |
| | Model 1 | | Model 2 | | Model 3 | | Model 4 | | Model 5 | | Model 6 | |
| | OR | 95% CI | OR | 95% CI | OR | 95% CI | OR | 95% CI | OR | 95% CI | OR | 95% CI |
|---|---|---|---|---|---|---|---|---|---|---|---|---|
| **Religion** | | | | | | | | | | | | |
| Muslim | | | 1.00 | | | | 1.00 | | | | 1.00 | | |
| Catholic | | | 0.74 | 0.54 1.01 | | | 0.74 | 0.54 1.03 | | | 0.73 | 0.53 1.00 |
| Zion | | | 0.36*** | 0.22 0.59 | | | 0.36*** | 0.22 0.59 | | | 0.35*** | 0.21 0.56 |
| Evangelical/Pentecostal | | | 0.47*** | 0.33 0.68 | | | 0.47*** | 0.32 0.67 | | | 0.47*** | 0.33 0.68 |
| No religion | | | 0.55* | 0.34 0.89 | | | 0.56* | 0.35 0.92 | | | 0.56* | 0.35 0.90 |
| Other | | | 0.29* | 0.11 0.77 | | | 0.30* | 0.12 0.79 | | | 0.30* | 0.11 0.78 |
| **Martial status** | | | | | | | | | | | | |
| Married | | | 1.00 | | | | 1.00 | | | | 1.00 | | |
| Living with partner | | | 0.79 | 0.57 1.08 | | | 0.73 | 0.53 1.00 | | | 0.79 | 0.58 1.09 |
| **Total children ever born** | | | | | | | | | | | | |
| None | | | 1.00 | | | | 1.00 | | | | 1.00 | | |
| One | | | 1.00 | 0.63 1.56 | | | 1.00 | 0.64 1.57 | | | 1.00 | 0.64 1.57 |
| Two | | | 0.87 | 0.54 1.40 | | | 0.87 | 0.54 1.40 | | | 0.87 | 0.54 1.40 |
| Three | | | 0.78 | 0.47 1.29 | | | 0.78 | 0.47 1.30 | | | 0.77 | 0.46 1.29 |
| Four or more | | | 0.62* | 0.40 0.98 | | | 0.62* | 0.39 0.97 | | | 0.62* | 0.39 0.98 |
| F | 23.66*** | | 7.23*** | | 15.38*** | | 5.99*** | | 23.24*** | | 7.86*** | |

*Note:* *p < 0.05, **p < 0.01, ***p < 0.001.

**Table 3.** Logit models predicting GAD-7 among married women in Mozambique

| | Respondent's health care | | | | Large household purchases | | | | Visits to family or relatives | | | |
| | Model 1 | | Model 2 | | Model 3 | | Model 4 | | Model 5 | | Model 6 | |
| | OR | 95% CI | OR | 95% CI | OR | 95% CI | OR | 95% CI | OR | 95% CI | OR | 95% CI |
|---|---|---|---|---|---|---|---|---|---|---|---|---|
| **Autonomy** | | | | | | | | | | | | |
| Partner/other | 1.00 | | 1.00 | | 1.00 | | 1.00 | | 1.00 | | 1.00 | |
| Respondent and partner | 0.77 | 0.56  1.05 | 0.93 | 0.67  1.29 | 0.73* | 0.54  0.98 | 0.83 | 0.60  1.13 | 0.81 | 0.60  1.08 | 0.92 | 0.67  1.26 |
| Respondent alone | 0.41*** | 0.31  0.56 | 0.52*** | 0.38  0.70 | 0.43*** | 0.29  0.64 | 0.52*** | 0.35  0.76 | 0.53*** | 0.38  0.75 | 0.64*** | 0.46  0.89 |
| **Education** | | | | | | | | | | | | |
| Secondary education or higher | | | 1.00 | | | | 1.00 | | | | 1.00 | |
| No education | | | 0.73 | 0.52  1.04 | | | 0.73 | 0.51  1.03 | | | 0.74 | 0.52  1.04 |
| Primary education | | | 1.06 | 0.81  1.40 | | | 1.06 | 0.81  1.39 | | | 1.08 | 0.83  1.42 |
| **Household wealth** | | | | | | | | | | | | |
| Poorer | | | 1.00 | | | | 1.00 | | | | 1.00 | |
| Poorest | | | 0.63** | 0.46  0.86 | | | 0.64** | 0.47  0.88 | | | 0.64** | 0.46  0.87 |
| Middle | | | 0.61* | 0.41  0.91 | | | 0.62* | 0.42  0.93 | | | 0.62* | 0.41  0.92 |
| Richer | | | 0.58** | 0.39  0.88 | | | 0.60* | 0.40  0.89 | | | 0.59** | 0.39  0.88 |
| Richest | | | 0.38*** | 0.23  0.63 | | | 0.39*** | 0.24  0.65 | | | 0.37*** | 0.22  0.62 |
| **Employment** | | | | | | | | | | | | |
| No | | | 1.00 | | | | 1.00 | | | | 1.00 | |
| Yes | | | 0.76 | 0.58  1.00 | | | 0.75* | 0.58  0.99 | | | 0.76* | 0.58  0.99 |
| **Age** | | | | | | | | | | | | |
| 15–19 | | | 1.00 | | | | 1.00 | | | | 1.00 | |
| 20–24 | | | 1.36 | 0.94  1.97 | | | 1.38 | 0.96  1.99 | | | 1.37 | 0.95  1.98 |
| 25–29 | | | 1.70** | 1.13  2.54 | | | 1.74** | 1.16  2.61 | | | 1.70** | 1.14  2.55 |
| 30–34 | | | 2.09** | 1.30  3.36 | | | 2.18*** | 1.35  3.51 | | | 2.09** | 1.30  3.36 |
| 35–39 | | | 2.79*** | 1.80  4.32 | | | 2.88*** | 1.85  4.47 | | | 2.76*** | 1.78  4.29 |
| 40–44 | | | 1.92** | 1.18  3.13 | | | 2.04** | 1.24  3.35 | | | 1.94** | 1.19  3.16 |
| 45–49 | | | 2.64*** | 1.65  4.22 | | | 2.76*** | 1.72  4.44 | | | 2.66*** | 1.66  4.27 |
| **Place of residence** | | | | | | | | | | | | |
| Urban | | | 1.00 | | | | 1.00 | | | | 1.00 | |
| Rural | | | 0.68 | 0.46  1.01 | | | 0.68 | 0.46  1.02 | | | 0.68 | 0.46  1.01 |

(*Continued*)

| | Respondent's health care | | | | Large household purchases | | | | Visits to family or relatives | | | |
|---|---|---|---|---|---|---|---|---|---|---|---|---|
| | Model 1 | | Model 2 | | Model 3 | | Model 4 | | Model 5 | | Model 6 | |
| | OR | 95% CI | OR | 95% CI | OR | 95% CI | OR | 95% CI | OR | 95% CI | OR | 95% CI |
| **Religion** | | | | | | | | | | | | |
| Muslim | | | 1.00 | | | | 1.00 | | | | 1.00 | |
| Catholic | | | 0.69* | 0.51 0.93 | | | 0.69* | 0.51 0.94 | | | 0.68* | 0.50 0.92 |
| Zion | | | 0.31*** | 0.20 0.47 | | | 0.30*** | 0.20 0.47 | | | 0.29*** | 0.19 0.45 |
| Evangelical/Pentecostal | | | 0.39*** | 0.25 0.59 | | | 0.38*** | 0.25 0.58 | | | 0.38*** | 0.25 0.58 |
| No religion | | | 0.55* | 0.33 0.91 | | | 0.56* | 0.34 0.93 | | | 0.55* | 0.33 0.92 |
| Other | | | 0.41* | 0.19 0.91 | | | 0.43* | 0.19 0.94 | | | 0.42* | 0.19 0.92 |
| **Martial status** | | | | | | | | | | | | |
| Married | | | 1.00 | | | | 1.00 | | | | 1.00 | |
| Living with partner | | | 0.81 | 0.64 1.04 | | | 0.77* | 0.60 0.98 | | | 0.79 | 0.62 1.01 |
| **Total children ever born** | | | | | | | | | | | | |
| None | | | 1.00 | | | | 1.00 | | | | 1.00 | |
| One | | | 0.90 | 0.59 1.38 | | | 0.91 | 0.59 1.39 | | | 0.91 | 0.60 1.40 |
| Two | | | 0.65 | 0.41 1.03 | | | 0.66 | 0.42 1.04 | | | 0.66 | 0.42 1.05 |
| Three | | | 0.69 | 0.43 1.11 | | | 0.69 | 0.43 1.11 | | | 0.70 | 0.43 1.13 |
| Four or more | | | 0.46*** | 0.30 0.71 | | | 0.46*** | 0.30 0.70 | | | 0.47*** | 0.30 0.72 |
| F | 18.00*** | | 7.20*** | | 9.17*** | | 5.98*** | | 6.96*** | | 5.77*** | |

*Note:* *$p < 0.05$, **$p < 0.01$, ***$p < 0.001$.

autonomy was limited to three questions related to women making decisions about their own health care, making large household purchases and their ability to visit family or relatives. Future research could employ a longitudinal approach focused on all women in Mozambique. Despite these limitations, however, our study is among the first to examine the association between women's household decision-making autonomy with depression and anxiety. It makes an important contribution to Mozambique's mental health policy.

Based on our findings, we have made some policy recommendations. First, given the observed association between women's household decision-making autonomy and poor mental health outcomes, it will be critical for health policymakers in our study context to design a holistic approach to mental health interventions among women. It would be crucial to identify the sociocultural factors working against women's autonomy and design specific responses to address them. Making men part of such intervention dialogues would work to increase their understanding of the adverse impact of women's poor autonomy on their mental health. Overall, increasing women's socioeconomic status, such as income and opportunities for employment, will not only lead to a better mental health outcomes but will also serve as an important pathway to increasing their autonomy. Building partnerships with religious groups would be useful in targeting especially Muslim women with the information and resources available for improved mental health.

**Open peer review.** To view the open peer review materials for this article, please visit http://doi.org/10.1017/gmh.2025.29.

**Data availability statement.** Data for this study is publicly available.

**Acknowledgements.** The authors are grateful to the Mozambique Demographic and Health Survey (DHS) for making the data available for this study.

**Author contribution.** Roger Antabe and Yujiro Sano conceptualized and designed the study. Gregory Antabe and Cornelius K. A. Pienaah coordinated administratively and conducted statistical analysis for the study and the investigations. Roger Antabe, Yujiro Sano and Gregory Antabe designed the methodology and statistical analysis. Roger Antabe, Gregory Antabe and Yujiro Sano wrote the first draft of the manuscript, and all authors contributed to and approved the final manuscript.

**Financial support.** This study did not receive any financial support.

**Competing interests.** The authors declare that they have no known competing financial interests or personal relationships that could have appeared to influence the work reported in this paper.

**Ethics statement (if appropriate).** All procedures performed in studies involving human participants were in accordance with the ethical standards of the institutional and/or national research committee and with the 1964 Helsinki Declaration and its later amendments or comparable ethical standards. The protocol for the MDHS was approved by the Instituto Nacional de Estatística Maputo, Moçambique and the Institutional Review Board of ICF, Rockville, Maryland, USA.

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
