## [Reviewer Report]

This paper examined women’s Household Decision-Making Autonomy and Mental Health Outcomes in Mozambique. The manuscript is well put together and will significantly contribute to the mental health field.

---

## [Reviewer Report]

The article appears to be well-structured with a comprehensive literature review, methodology, data analysis, and discussion of findings. However, to enhance its clarity and impact, consider the following suggestions:

1. Introduction Section

- Summarize the primary research gaps or why this study is unique and necessary, as this can enhance the justification for the study.

2. Literature Review

- Although comprehensive, you could highlight more clearly what makes this study different. For example, women’s exposure to gender roles is a major obstacle to decision-making, which can negatively impact their mental health.

(https://doi.org/10.35365/ctjpp.21.3.22)

- Include recent studies on women’s self-perception, depression, and anxiety levels to ensure it is up-to-date.

(https://doi.org/10.1007/s11469-022-00910-1)

3. Methodology

- Please add a brief explanation of each scale within the methodology for clarity.

4. Discussion

- Discuss potential limitations of the study (e.g., cross-sectional design) and suggest directions for future longitudinal research.

6. Conclusion and Recommendations

- Summarize the key findings more succinctly, focusing on direct implications for clinical practice and policy.

---

## [Reviewer Report]

congratulate the authors on the topic, which is very pertinent to the Mozambican context and there are few studies.

Recommendations:

1. Explore a little more about the psychosocial factors of depression to

enrich the discussion of findings.

2. Search for articles by Massinga Luciana, Mandlate Flavio, Levero Kate, Wainberg Milton, Mootz Jennifer and Santos Palmira.

---

## [Editor Report]

• SSA is used, but not defined. I suppose it refers to “Sub-saharan Africa). If so, I suggest using instead more geographically accurate terms (i.e., easter, southern, etc.) or even the names of specific countries. 

• Please, include both, the number of people (N) and the percentage (%) in Table 1. 

• Data analysis is sound. Although a sentence or two about the National survey could be helpful for unfamiliar readers (i.e. is it a phone survey?, individual or group interview?, etc.)

• Because policy recommendations are part of the study objective, this section could benefit from more specificity and elaboration.

• Women’s autonomy was only asked among women who are married and living with their partners, thereby excluding unmarried, divorced, and widowed women. This is a study limitation that is not mentioned.

• Level of severity of depressive and anxiety symptoms are not examined, which seems like another important limitation given its impact on autonomy and agency. However, it is something the study could potentially answer, especially as it may lead to more specific and nuanced policy recommendations, increasing the impact of the study.